# Structurally robust lithium-rich layered oxides for high-energy and long-lasting cathodes

Ho-Young Jang[1,11], Donggun Eum [1,2,10,11], Jiung Cho[3], Jun Lim [4], Yeji Lee[4], Jun-Hyuk Song[1], Hyeokjun Park[5], Byunghoon Kim [1,2], Do-Hoon Kim[1], Sung-Pyo Cho[6], Sugeun Jo [7], Jae Hoon Heo [1], Sunyoung Lee [1], Jongwoo Lim [7] & Kisuk Kang [1,2,8,9] ✉

O2-type lithium-rich layered oxides, known for mitigating irreversible transition metal migration and voltage decay, provide suitable framework for exploring the inherent properties of oxygen redox. Here, we present a series of O2-type lithium-rich layered oxides exhibiting minimal structural disordering and stable voltage retention even with high anionic redox participation based on the nominal composition. Notably, we observe a distinct asymmetric lattice breathing phenomenon within the layered framework driven by excessive oxygen redox, which includes substantial particle-level mechanical stress and the microcracks formation during cycling. This chemo-mechanical degradation can be effectively mitigated by balancing the anionic and cationic redox capabilities, securing both high discharge voltage (~ 3.43 V $vs$. Li/Li$^+$) and capacity (~ 200 mAh g$^{-1}$) over extended cycles. The observed correlation between the oxygen redox capability and the structural evolution of the layered framework suggests the distinct intrinsic capacity fading mechanism that differs from the previously proposed voltage fading mode.

Lithium-rich layered oxides (LRLOs) are highly regarded as one of the potential cathode materials for next-generation high-energy lithium batteries offering both the economic superiority from earth-abundant Mn-rich compositions and high theoretical energy density owing to the anionic redox activation[1–3]. While traditional cathodes in lithium batteries mostly rely on redox activity of transition metal for electrochemical reaction, LRLOs exploit redox activities of both oxygen and transition metal cumulatively, thus being capable of delivering an exceptionally high capacity that conventional cathode materials have not been able to offer[1,4–7]. The unique capability of the cumulative redox activity is ascribed to the presence of residual lithium ions in transition-metal layers, i.e., excess lithium content in a layered

[1]Department of Materials Science and Engineering, Institute for Rechargeable Battery Innovations, Research Institute of Advanced Materials, Seoul National University, 1 Gwanak-ro, Gwanak-gu, Seoul 08826, Republic of Korea. [2]Center for Nanoparticle Research, Institute for Basic Science (IBS), Seoul National University, 1 Gwanak-ro, Gwanak-gu, Seoul 08826, Republic of Korea. [3]Seoul Western Center, Korea Basic Science Institute (KBSI), 150 Bugahyeon-ro, Seodaemun-gu, Seoul 03759, Republic of Korea. [4]Pohang Light Source-II, Pohang University of Science and Technology (POSTECH), 80 Jigok-ro 127 beon-gil, Nam-gu, Pohang 36763, Republic of Korea. [5]Interdisciplinary Materials Measurement Institute, Korea Research Institute of Standards and Science (KRISS), Daejeon 34113, Republic of Korea. [6]National Center for Inter-University Research Facilities, Seoul National University, 1 Gwanak-ro, Gwanak-gu, Seoul 08826, Republic of Korea. [7]Department of Chemistry, College of Science, Seoul National University, 1 Gwanak-ro, Gwanak-gu, Seoul 08826, Republic of Korea. [8]Institute of Engineering Research, College of Engineering, Seoul National University, Seoul 08826, Republic of Korea. [9]School of Chemical and Biological Engineering, College of Engineering, Seoul National University, 1 Gwanak-ro, Gwanak-gu, Seoul 08826, Republic of Korea. [10]Present address: Department of Materials Science and Engineering, Stanford University, Stanford, CA, USA. [11]These authors contributed equally: Ho-Young Jang, Donggun Eum. ✉e-mail: matlgen1@snu.ac.kr

structure. It creates new localized oxygen electronic states at a relatively high energy level, thereby activating lattice oxygen redox activity in the accessible voltage range[4,8,9]. Despite their promise, one significant issue with LRLO cathodes has been the unremitting decay of operating voltage, hampering their practical use[10–12]. Conventional LRLOs typically suffer from a gradual decline in discharge voltage and a loss of the intrinsic redox properties over cycles, which limit the advantages of their high energy density. Researchers have spent decades investigating the cause of voltage decay and have found that it may be due to the migration of transition metals within the layered structure degrading the original layered structure to a partially disordered oxide, which causes significant and irreversible changes in oxygen redox activity and stability[6,10,13,14].

Recent experiments have shown that restricting transition metal migration in a crystal can significantly reduce the voltage decay of LRLOs[13]. This was achieved by altering the stacking sequences of oxygen layers from conventional O3-type to O2-type stacked layered structures[13]. In the O2-type LRLOs, the transition metal migrations are locked in a specific direction provided by the O2-type oxygen local environments, and these structural features could inhibit the accumulation of displaced transition metals and the corresponding formation of disordered-phase during electrochemical cycling[6,8,13–16]. Consequently, it was shown that a stable voltage retention, i.e., long-lasting bottleneck of LRLO cathode, was attainable over prolonged cycling, presenting O2-type LRLOs as a new class of electrode materials for reversible anionic redox reaction. And, more recently, Liu et al. demonstrated that partial occupancy of transition metals in lithium layer can form capped-honeycomb structure in O2-type LRLOs, preserving the high-voltage stability[17]. These studies imply that the O2-LRLO can be an appropriate platform to elucidate the intrinsic property of oxygen redox reaction in layered oxides, decoupled from extrinsic factors such as disordering of the pristine structure. Nevertheless, our understanding on these new electrode materials is still in its infancy, especially regarding the extent of the anionic redox capability and its relation with the structural stability. For example, it remains elusive how far we can go with the oxygen redox in the layered oxide framework without structural instability caused by highly oxidized lattice oxygen[15]. Furthermore, it is not certain whether the reversibility of full oxygen redox activity, if achievable in layered oxides, translates to the electrode-level stability and reversibility in a practically long-term cycling.

Here, we aim to address these fundamental questions surrounding the oxygen redox activity taking the intact layered framework of O2-LRLOs as a platform. Our series of O2-LRLOs with varying Li concentrations was adopted in order to diversify the relative portion of cationic and anionic redox, which consequently results in the more excessive Li-containing materials than the typical LRLO materials[18,19]. We demonstrate that the O2-LRLO cathodes are capable of serving as a stable oxygen redox framework that can utilize the anionic redox reaction as much as ~90% based on the nominal composition without any voltage fade or transition metal disordering over electrochemical cycles. Moreover, our comparative analysis on these newly developed electrodes, which exhibit varying oxygen redox capabilities, unveils a unique and intrinsic correlation between the extent of the oxygen redox and lattice breathing of the layered framework. This crucial relation remained elusive in conventional LRLOs due to their inherent structural disordering. We further show that this connection between high oxygen redox activity and the large lattice variation can lead to particle-level mechanical crack formations (and consequent capacity degradation) as cycling progresses. It bears interesting resemblance with the chemo-mechanical failure mode of nickel-rich layered oxide cathodes[20–22], manifesting the shared challenges that need to be overcome in high-capacity layered oxide cathodes. Finally, it is demonstrated that the chemo-mechanical degradation in LRLO electrodes can be substantially mitigated by balancing anionic and cationic redox capabilities, showcasing an O2-LRLO cathode that can successfully retain both high voltage (~3.43 V vs. Li/Li$^+$) and capacity (~220 mAh g$^{-1}$) over extended cycles, an achievement that has remained elusive with conventional LRLO electrodes.

## Results

### Degree of oxygen redox activities in O2-LRLOs and its consequences

We prepared O2-type LRLO samples with five different compositions by controlling the excess amount of lithium, a key enabler of oxygen redox activity in LRLO[8], to comparatively probe the extent of the anionic redox capability. The targeted compositions of O2-type LRLOs were $Li_x(Li_yNi_{(5-18y)/10}Mn_{(8y+5)/10})O_2$ ($x \approx 0.83$, see a compositional diagram in Supplementary Fig. S1), where $y$ corresponds to lithium content in the transition metal layer, increasing from 0.15 to 0.25 in increments of 0.025. (Sample names are hereafter designated as $LL_y$.) This series of O2-type LRLOs were synthesized via ion exchange process from the equivalent P2-type sodium layered materials (Supplementary Fig. S2 and Fig. S2a)[13,23]. High-resolution powder diffraction (HRPD) patterns in Supplementary Fig. S3b indicate that major peak positions coincide with the characteristic Bragg positions of the O2-type layered structures with $P6_3mc$ space group along with superstructure peaks that appear at $2\theta = 20 - 30°$, suggesting a successful synthesis of O2-LRLOs[13,24,25]. Despite the detection of minor O3-type $Li_2MnO_3$ impurities in both P2- and O2-structures (Supplementary Fig. S4), the marginal activity of the impurity phase (Supplementary Fig. S5–6) assures that the primary O2-phase dominates the electrochemical and structural responses. All the $LL_y$s exhibited similar single-crystalline, platelet particle morphology with around 4μm of particle size (Supplementary Figs. S7–8). X-ray absorption near-edge spectra (XANES) display that the half-edge energy positions of the Ni K-edge are similarly superimposed for all five samples, which are close to that of the reference material of Ni$^{2+}$O (Supplementary Fig. S9a). The Mn K-edge energies are also nearly identical for all the samples and comparable to that of the Mn$^{4+}$O$_2$ reference in Supplementary Fig. S9b. These XANES results collectively suggest that all the pristine LRLOs possess similar Ni and Mn oxidation states, estimated as ~2.33+ and 4+ judged from the nominal compositions, respectively. See more details on the sample analysis in Supplementary Note 1 and Table S1.

Different lithium excess levels in the five O2-LRLOs are supposed to subsequently exhibit varying degrees of anionic (or oxygen) redox activities. The Li–O–Li configuration resulting from the excess lithium in LRLO produces unhybridized oxygen states, which is widely recognized to contribute to the anionic redox reaction[4–6], as depicted with O$^{(II)}$–Li$_4$TM$_2$ in Fig. 1a. As more transition metal ions are replaced by excess lithium ions in LRLOs from LL$_{0.15}$ to LL$_{0.25}$, the oxygen configuration of the O$^{(II)}$ type becomes more dominant, as presented in Fig. 1b. Accordingly, a growing extent of oxygen redox activity is expected from LL$_{0.15}$ to LL$_{0.25}$, which would possess approximately 45–75% O$^{(II)}$ type oxygen in the overall oxygen content. Considering the initial oxidation state of transition metal and the nominal compositions of O2-LRLOs, the cationic redox reaction (i.e., Ni$^{2+}$/Ni$^{3+}$/Ni$^{4+}$) decreases from ~39% in LL$_{0.15}$ to ~7% in LL$_{0.25}$ (see Supplementary Fig. S10b for the estimation), indicating that LL$_{0.25}$ sample would undergo the oxygen redox as much as ~92% of the total redox reaction in the electrochemical reaction. To corroborate these estimations, we comparatively measured the changes in the O K-edge when charging each LRLO electrode to 4.6 V by scanning transmission X-ray microscopy (STXM) analysis[26,27] as presented in Supplementary Fig. S11. It clearly indicates that the anionic redox becomes a far more dominating reaction over the cationic redox as the extent of lithium excess increases from LL$_{0.15}$ to LL$_{0.25}$ (see Supplementary Note 2 for details).

The electrochemical properties were comparatively examined for O2-LRLO electrodes in lithium electrochemical cells, as depicted in

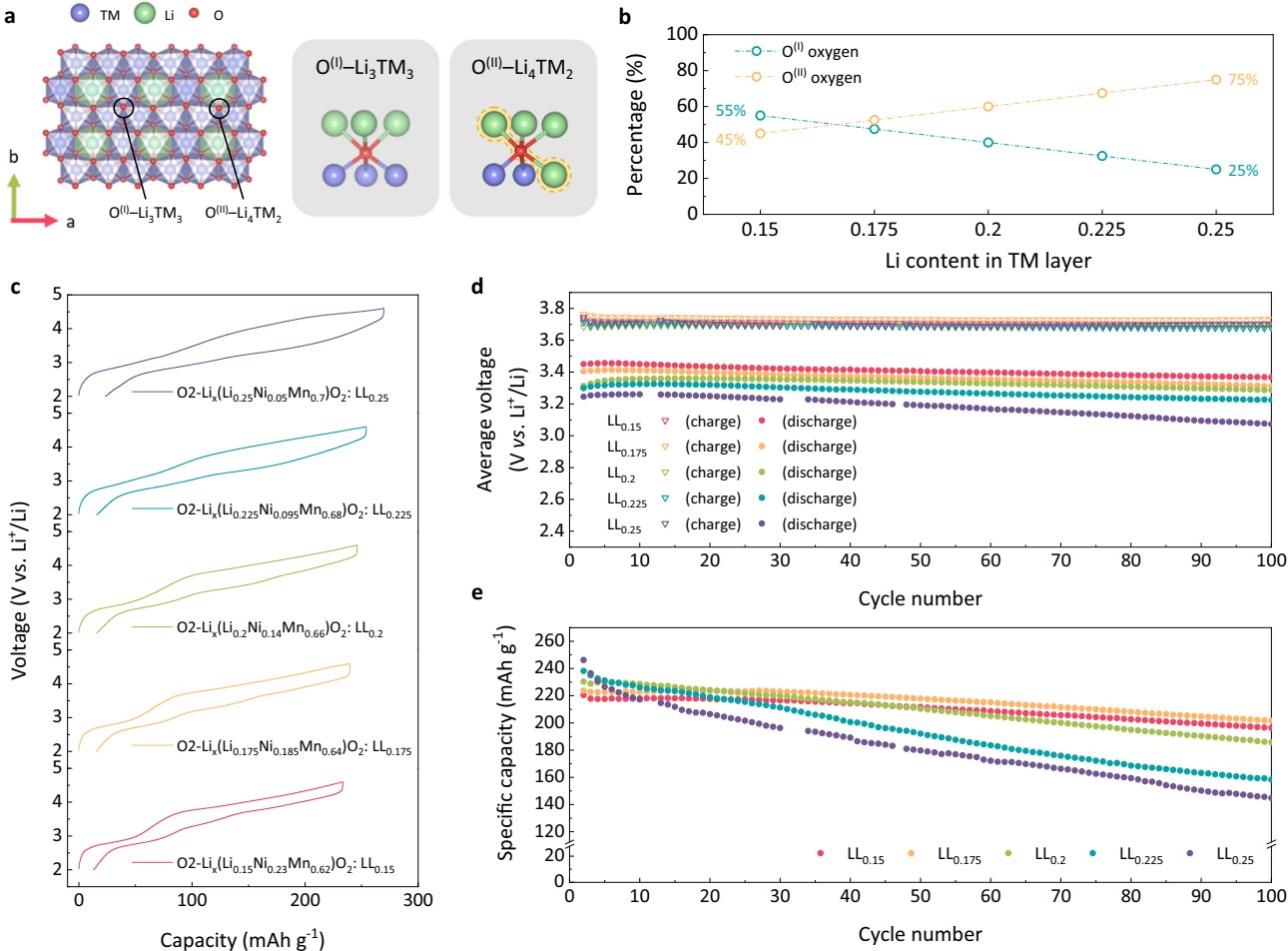

**Fig. 1 | Comparison of electrochemical properties of O2-type LRLOs.**
**a** Schematic illustration of the transition metal (TM) layer and oxygen types in typical LRLOs. O$^{(I)}$ and O$^{(II)}$ are magnified in right figures, indicating the oxygen species coordinated with 3Li–3TM and 4Li-2TM, respectively. The Li-O-Li local configuration in 4Li-2TM is denoted with yellow dotted line. **b** Theoretical relative fraction of the O$^{(I)}$ and O$^{(II)}$ in (**a**) as a function of the excess Li content in TM layer. **c** Second charge–discharge curves of LL$_y$s cycled within 2.0–4.6 V voltage range at a current density of 20 mAg$^{-1}$, at 60 °C. **d, e** Average charge and discharge voltages and specific discharge capacity for all the LL$_y$s for 100 cycles measured at a current density of 20 mA g$^{-1}$, at 60 °C.

Fig. 1c–f. Figure 1c displays the charge and discharge profiles of each electrode in the second cycle after the initial formation cycle (Supplementary Fig. S10a). As the excess lithium content increases, a higher specific capacity could be attained, which is partly due to lighter molecular weights of higher-lithium excess samples. LL$_{0.15}$, LL$_{0.175}$, LL$_{0.2}$, LL$_{0.225}$ and LL$_{0.25}$ electrodes could deliver discharge capacities of 220, 224, 230, 238 and 246 mAh g$^{-1}$, respectively, at a current density of 20 mA g$^{-1}$ at 60 °C, corresponding to 0.71, 0.71, 0.72, 0.73, and 0.74 lithium utilizations per formula unit (Supplementary Fig. S12). It indicates that comparable amounts of lithium ions participated in the electrochemical reaction regardless of the initial compositions of the samples and the corresponding oxygen redox contribution. On the other hand, the average discharge voltage gradually decreases from LL$_{0.15}$ to LL$_{0.25}$ electrodes, as presented in Fig. 1d. The discharge voltage of LL$_{0.15}$ is the highest with 3.43 V (*vs.* Li/Li$^+$), which systematically declines to 3.40, 3.36, 3.29 and 3.23 V for LL$_{0.175}$, LL$_{0.2}$, LL$_{0.225}$ and LL$_{0.25}$ electrode, respectively. It specifies the inverse correlation between the average discharge voltage and the degree of oxygen-redox involvement in O2-LRLOs electrodes. Interestingly, we further found that the average charge voltages appear to be almost identical for all the O2-LRLO electrodes, revealing a higher voltage hysteresis for electrodes with a greater anionic redox contribution (more quantitative plots are provided in Supplementary Fig. S13). It clearly demonstrates that oxygen redox reactions are inherently more sluggish than those of

cationic counterpart, which is the topic that needs further fundamental studies[26,28–31]. Nevertheless, we note that all the O2-LRLOs in this study exhibited negligible voltage decay over electrochemical cycling (<0.05 V for 50 cycles), confirming the universal efficacy of O2-type layered materials against the voltage decay in LRLOs[13,23].

Figure 1e illustrates that all the electrodes could retain respectable cycle stability, exhibiting 89%, 90%, 81%, 66% and 59% of capacity retentions after 100 cycles for LL$_{0.15}$, LL$_{0.175}$, LL$_{0.2}$, LL$_{0.225}$ and LL$_{0.25}$ electrode, respectively. Notably, the LL$_{0.25}$ electrode, which contains only 5% Ni in the transition metal layer of the Li$_x$(Li$_{0.25}$Ni$_{0.05}$Mn$_{0.7}$)O$_2$ formula, exhibits impressive cycle performance. Given that approximately 92% of the total redox reaction is attributed to O$^{2-}$/O$^-$ activity based on the nominal composition, this is particularly noteworthy. It is a stark contrast to the equivalent O3-type Li(Li$_{1/3}$Mn$_{2/3}$)O$_2$ electrode, which mainly undergoes the oxygen redox reaction but barely retains its redox stability over cycles[32–34]. Moreover, LL$_{0.225}$ which contains only 9.5% Ni, i.e., Li$_x$(Li$_{0.225}$Ni$_{0.095}$Mn$_{0.68}$)O$_2$, also presents remarkable cycle stability while displaying the robust voltage retention over cycles. Benefiting both from the voltage stability and the decent capacity retention, the specific energy densities of O2-LRLO electrodes remained markedly stable over cycles, delivering as much as 660.9 Wh kg$^{-1}$ after 100 cycles in LL$_{0.15}$, as shown in Supplementary Fig. S14. Nonetheless, we acknowledge that the capacity retention properties differ noticeably depending on *y* values in LL$_y$ electrodes

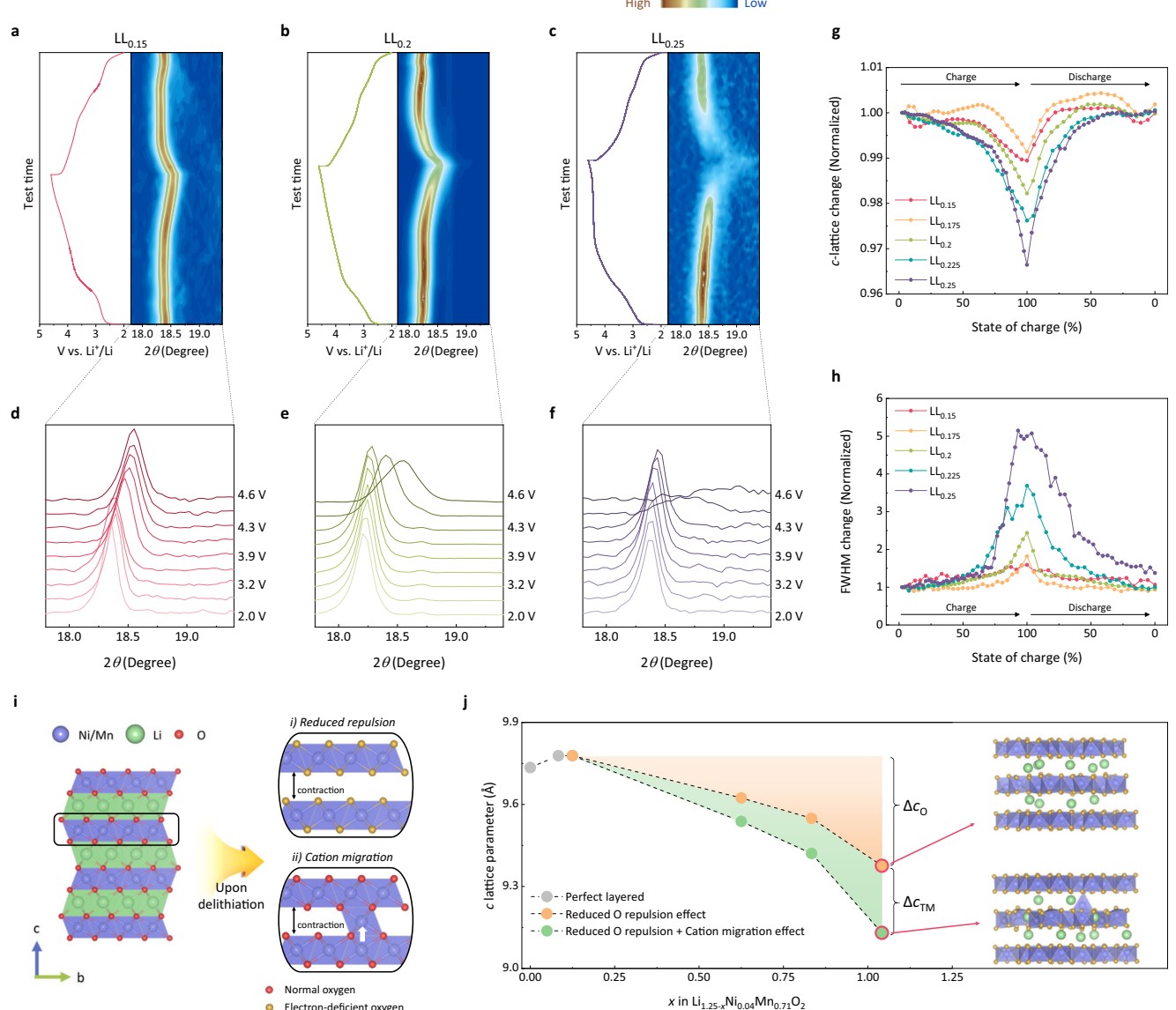

**Fig. 2 | Lattice evolution induced by anionic redox activation in O2-type LRLOs.** Second-cycle charge–discharge profiles and corresponding in situ diffraction patterns (right) for LL$_{0.15}$ (**a**), LL$_{0.2}$ (**b**), and LL$_{0.25}$ (**c**). All the cycles were performed within the voltage window of 2.0–4.6 V and at a current density of 20 mA g$^{-1}$. **d–f**, Magnified (002) XRD peak of LL$_{0.15}$ (**d**), LL$_{0.2}$ (**e**), and LL$_{0.25}$ (**f**) during the charging process. Comparison of normalized $c$-lattice parameter evolution (**g**) and FWHM changes (**h**) of (002) Bragg peak (**e**) for LL$_y$ electrodes. **i** Two main factors that can influence the shrinkage of the layered structure during high-voltage anion redox reactions: (i) reduced repulsion between electron-dissipated anion frameworks and

(ii) cation migration into the vacant interlayer. **j** Theoretical $c$-lattice evolution trend of computationally modeled LL$_{0.25}$ electrode. The $c$-lattice parameters for simple oxygen-oxidated structures (orange) and structures with additional TM-migration (green) at the various SOCs in comparisons with the reference layered structures (gray). The amount of lattice change resulted from each oxygen repulsion reduction and TM migration ($\Delta c_O$ and $\Delta c_{TM}$, respectively) was presented with orange and green shade, respectively, while the final calculated structures were shown in the right.

unlike the commonly observed voltage stability feature. The LL$_{0.15}$ and LL$_{0.175}$ electrodes maintain close to ~ 90% of the initial capacity after cycles, while the capacity retentions of LL$_{0.225}$ and LL$_{0.25}$ electrodes with relatively greater anionic redox contributions were comparatively lower. Furthermore, at high current density of 1000 mA g$^{-1}$, the single-crystalline LL$_{0.15}$ and LL$_{0.2}$ retained 64.5 and 56.1% of their initial capacity, respectively, which is comparable or even superior to poly-crystalline conventional O3-type LRLO materials[35–37] (Supplementary Fig. S15). However, in the same current density, LL$_{0.25}$ could only deliver 0.36% of its original capacity, indicating severe kinetic limitation associated with anionic redox capability. These observations strongly suggest that the capacity fading mode in O2-LRLOs may not be linked to the voltage fade mechanism, which is known to be

associated with irreversible disorder in the layered structure during repeated anionic redox reaction[6,10,30,38,39].

## Distinct structural evolutions of layered structure upon oxygen redox

In order to understand this distinct behavior, we carefully monitored the structural change of each O2-LRLO electrode as a function of state-of-charges (SOCs) in an electrochemical cycle by in situ X-ray diffraction measurements. Figure 2a–f and Supplementary Fig. S16 depict the evolution of the main characteristic (002) peak of the O2-type crystalline phase for each electrode. It shows that the (002) diffraction peak is gradually shifted to a higher angle with charging to 4.6 V for all the electrodes. And, upon discharging to 2.0 V, the diffraction peak

nearly returns to its original position, indicating a reversible one-phase-based structural change with de-/lithiation. The reversible structural restoration could be also detected from Raman spectroscopy (see Supplementary Note 3 and Fig. S17), confirming the lattice structural reversibility of O2-type layered material even with a high level of oxygen redox reaction. However, noticeable differences were evident among electrodes especially at high SOCs, concerning the extent of peak shift and the peak intensity (Fig. 2a–c and Supplementary Fig. S16a-b). As more oxygen redox is involved from $LL_{0.15}$ to $LL_{0.25}$, the electrode displayed substantially greater shift of 2θ and more significant peak broadening at high SOCs, as closely displayed with the extended view in Fig. 2d–f and Supplementary Fig. S16c, d. When charging up to 4.3 V, all electrodes commonly showed continuous peak shift with similar degrees. On the other hand, the drastic change began to appear above 4.3 V particularly for the $LL_y$ electrode with higher $y$ values. Considering that the onset voltage of the oxygen redox is near 4.3 V ($vs.$ Li/Li+, Supplementary Fig. S18), it strongly suggests that the abrupt structural change is associated with the oxygen redox taking place in the O2-type layered structure.

In Fig. 2g, we quantitatively analyzed the $c$-axis lattice parameter to determine the extent of the lattice variation in each electrode. The figure shows that the $c$-lattice parameter shrinkage is markedly severe in the high SOC region, which is analogous to what is often observed in nickel-rich layered oxide cathodes[22,40,41]. Notably, the $LL_{0.25}$ electrode with the highest lithium ($y$) content exhibited the greatest shrinkage. The fully charged $LL_{0.25}$ underwent a $c$-lattice contraction of nearly 3.35% at the end of the charge, whereas 1.78% and 0.86% shrinkage were observed in the $LL_{0.2}$ and $LL_{0.15}$ electrodes, respectively. In the subsequent discharge process, the structural parameters returned to their original values, and these behaviors were similarly repeated in the next cycle (Supplementary Fig. S19). Unlike the $c$-lattice parameter, the $a$-lattice parameters presented negligible changes during de-/lithiation of electrodes (Supplementary Fig. S20), suggesting a large anisotropic lattice evolution especially for $LL_{0.25}$. We could also observe that the large lattice asymmetric change accompanies a significant peak broadening as evidenced from the changes in FWHM (full width at half maximum) values in Fig. 2h[42]. Consistent with the findings in Fig. 2g, the electrodes displaying the greatest lattice asymmetry exhibited the most substantial change in FWHM, i.e., $LL_{0.25}$ electrode. Since the asymmetric lattice change would accompany a significant stress/strain, repeated cycles of contraction and recovery would entail continuous strain buildup in the electrode particles[22,40,41].

According to previous studies on layered lithium transition metal oxides, two major factors have been widely known to influence the $c$-lattice collapse during the de-lithiation process: (i) reduced repulsion between electron-dissipated anion frameworks[43–45] and (ii) transition metal migration into the vacant lithium layer[15,46,47] (Fig. 2i). At a highly de-lithiated state, the electron-deficient O− state exerts weaker repulsive force between two adjacent oxygens and easily create dimerized oxygen pairs[48–50] compared with the pristine O2−, resulting in a comparably narrow and irregular slab distance between the layers. Similarly, a transition metal that has been displaced into the unoccupied lithium layer may be able to narrow the gap between two oxygen layers, owing to its smaller ionic radius. In order to clarify the origin of the abrupt $c$-lattice change, we computationally modeled the lattice evolution of the $LL_{0.25}$ upon charging and attempted to decouple the two factors, as presented in Fig. 2j. The figure shows that a drastic $c$-lattice parameter contraction takes place even without the transition metal migration (orange line, denoted with $\Delta c_O$), which accounts for 0.40 Å at the end of the charge. We could also observe that the transition metal migration, if it occurs, contributes to the lattice shrinkage additionally by approximately 0.24 Å as denoted with $\Delta c_{TM}$ in the figure. It clearly reveals that the contribution of the oxygen oxidation is more dominant in the total contraction of the $c$-lattice over the effect of the displaced transition metal. It is also consistent with our observations that O2-LRLO electrodes with a greater oxygen redox activity undergoes a larger degree of the $c$-lattice collapse and the asymmetry, implying a general challenge and trade-off between the stability and the high-capacity oxygen-redox in layered structure. (See Supplementary Note 3–4 and Figs. S16, S21–S22 for additional discussion about the structure variation, the oxygen redox and transition metal migration.)

## Chemo-mechanical degradation of anionic-redox electrodes

Such asymmetric contraction and expansion of the lattice parameter, which is presumably driven by the oxygen redox, would inevitably accompany a large volume change, thereby inducing severe mechanical stress at a particle level[40]. The stress can be accumulated with repeated cycling and lead to chemo-mechanical failure, causing premature capacity degradation. In order to verify this hypothesis, we conducted X-ray nano-imaging analysis on the cycled electrodes and assessed the particle integrity by analyzing over a thousand two-dimensional tomographic cross-sectional images. Figure 3a illustrates the representative images of cycled $LL_y$ electrodes within a window size of 40 μm × 40 μm. While micron-sized particles are readily identifiable in each tomographic image of the electrodes, a detailed analysis uncovered that a significant number of the particles were cracked. Interestingly, these cracks were more frequently observed with particles in $LL_y$ electrode with high $y$ values as indicated with yellow circles in each tomographic image. It is contrast to the case of the pristine electrodes that were nearly absent with such structural defects and exhibited contain comparable stress states (Supplementary Figs. S23–S24). In order to further clarify the nature of the cracks, we conducted three-dimensional rendering process to reconstruct a particle morphology in each cycled electrode (indicated by yellow arrows in Fig. 3a) in Fig. 3b. A sequence of slice images clearly shows that these cracks are internal cracks within the particles, as evinced by the dark ripped features. Furthermore, particles in cycled electrodes with greater oxygen redox activity contained a more extensive distribution of internal cracks. While premature crack formation was only slightly spotted in the case of $LL_{0.15}$, dark ripped features were growingly more discernable in the particles of $LL_{0.225}$ and $LL_{0.25}$ electrodes. In the case of $LL_{0.25}$ electrode, the ripped features evolved into sharp strips as marked with the red arrows in Fig. 3b, which is a typical feature of intragranular cracking[20,21]. We could roughly quantify the relative volume fraction of the damaged region in particles, i.e., the ratio of the crack volume to the total volume, as depicted in Fig. 3c. It exemplifies that the proportion of defect volume increases from 0.1 to 7.9% as the y in $LL_y$ rises from 0.15 to 0.25. For more statistical analysis, we randomly selected 10 particles from each electrode (Supplementary Fig. S25) and performed the same rendering process to average the defect volume ratio, as displayed in Fig. 3d. It confirmed that the crack volume ratio in the particles drastically increases from 0.06 to 7.35% for $LL_{0.15}$ and $LL_{0.25}$ electrodes, respectively. This result strongly supports the idea that the stress/strain buildup caused by the extensive utilization of high-voltage anionic redox entails severe chemo-mechanical failure, which can lead to more rapid capacity degradation as observed in Fig. 1e.

We further conducted scanning transmission electron microscopy (STEM) analysis on internal cracks of electrode particles. The low-magnification images in Fig. 4a reveal two-dimensional nature of intragranular cracks, suggesting that these cracks have propagated along the interslab space in the layered oxide[20,21]. Upon closer examination in Fig. 4b, we confirmed that the crack has propagated along the (002) plane in the O2-type layered structure. This atomic-level analysis also exposed that the presence of an internal crack is consistently accompanied by a 'near-crack region' with significant structural degradation that deviates from the pristine layered structure. The fast Fourier transformed (FFT) image of the near-crack region (or region II in Fig. 4b, d) exhibits a pattern that is considerably distinct

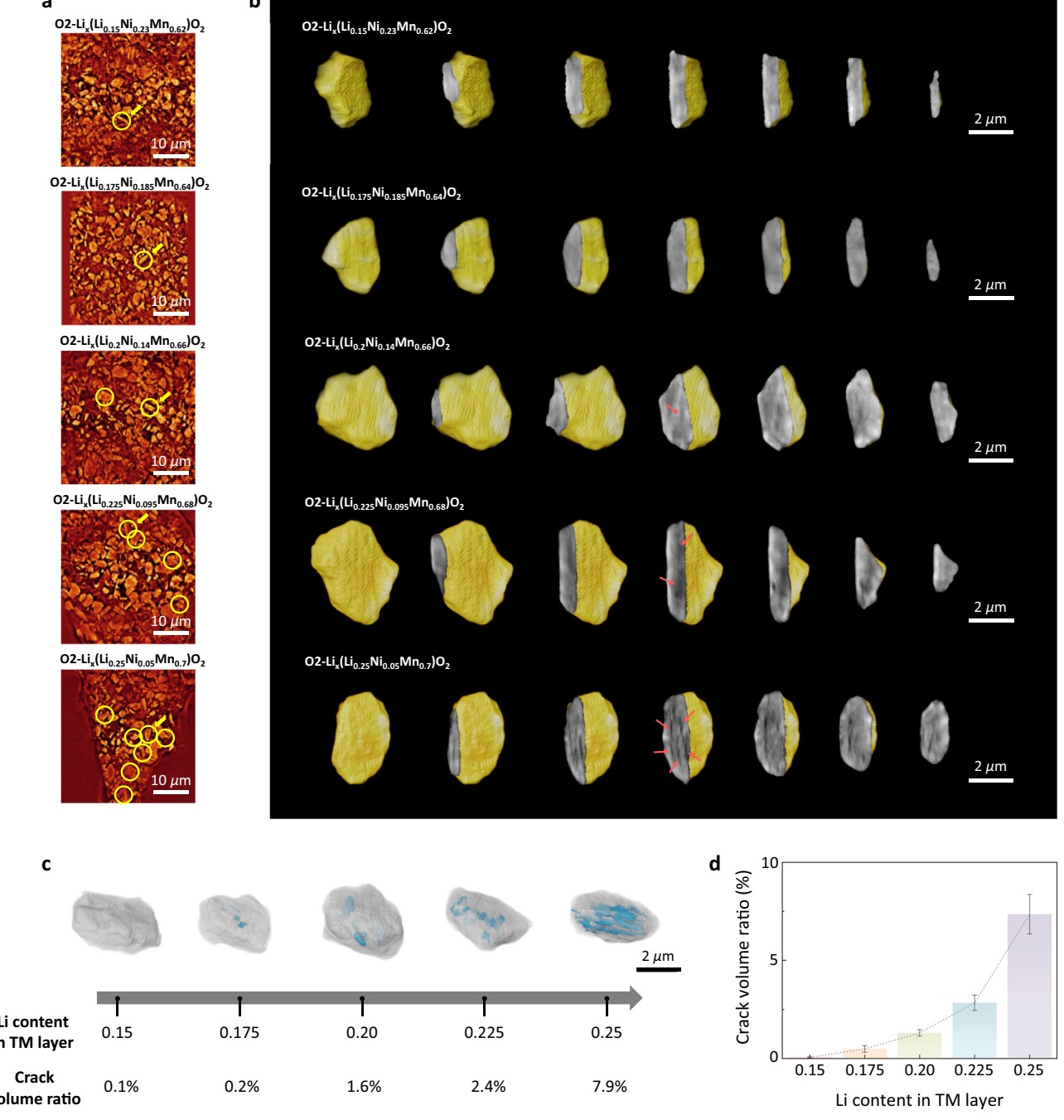

**Fig. 3 | Intragranular crack formation followed by anionic redox in O2-type LRLOs. a** XNI tomographic images inside the tenth-charged $LL_y$ electrodes. The particles with bulk defects are marked with yellow circles. **b** Three-dimensional volume rendered images and corresponding progressive cross-sectional views of one of the defective particles (designated with yellow arrow in (**a**)) in tenth-charged $LL_y$ electrodes. The dark strips indicated by the red arrows indicate intragranular cracks, which were especially conspicuous in $LL_{0.25}$. **c** Volume ratio of bulk defect region (blue) to the total particle (gray) with $Li_{TM}$ contents. (Each particle is that used in the volume rendition in (**b**)) **d** Trimmed-mean crack volume ratio values calculated for ten random particles of each $LL_y$ (see Supplementary Fig. 25).

from the characteristic spot patterns of the charged O2 layered structures[13] in Fig. 4c. Instead, the patterns in region II correspond to those of the $Pna2_1$ disordered $Mn_2O_3$ phase along the [010] zone axis (Supplementary Fig. S26). Signal profiles for the two regions in Fig. 4e, f also confirm that significant disorder has taken place in region II (near the cracks) with a notably reduced lattice spacing[13]. The near-crack region was additionally examined through chemical analysis using electron energy-loss spectroscopy (EELS) measurements in Fig. 4g–j. It shows that the pre-edge peak of the O K-edge (527.9–554.1 eV) is substantially depressed near the crack, whereas the Mn $L_{3,2}$-edges

(639.4–660.7 eV) shift to a lower-energy region (see Supplementary Fig. S27 for details). The relative oxygen content and oxidation state of Mn in Fig. 4i, j clearly depict that the average relative oxygen composition gradually decreases, while the $Mn^{4+}$ ions were reduced to $Mn^{(3+\delta)+}$ when approaching the crack, as evidenced by the change in Mn $L_3/L_2$ intensity ratio[51–53]. This series of observations serves to highlight a clear correlation between the formation of cracks and the transition to an electrochemically inactive disordered phase, providing compelling evidence in support of the chemo-mechanical degradation of O2-LRLO electrodes with a higher degree of oxygen redox participation.

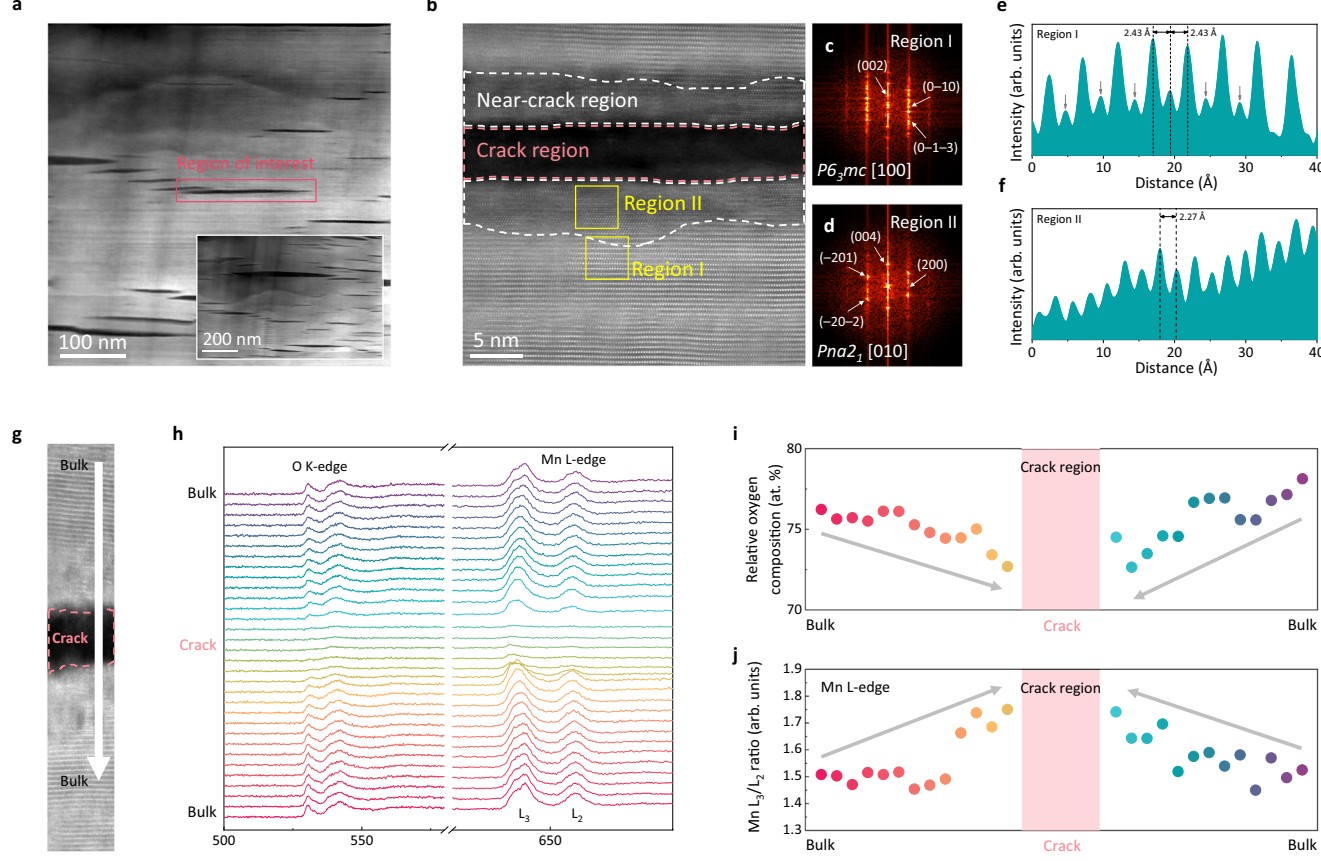

**Fig. 4 | Structural evolution with anionic-redox-driven crack formation. a** STEM image for tenth-charged LL$_{0.25}$ particle. The inset shows the entirely fractured features of the LL$_{0.25}$ particle. **b** Magnified image of one of the crack regions marked with the red box in (**a**). The crack region and surrounding area are enclosed with pink and white dashed lines, respectively. FFT patterns of the crack-free region (**c**, region I) and near-crack region (**d**, region II), surrounded by the yellow boxes in (**b**). Compared to the well-maintained *P6$_3$mc* structure in (**c**), characteristic planes for *Pna2$_1$* Mn$_2$O$_3$-like structure are indexed in (**d**) with white arrows. HAADF signal profiles for region I (far from crack) (**e**) and region II (near the crack) (**f**). The gray arrow in the signal profile in region I indicates the migrated TM into the Li layer. **g** STEM-EELS scan direction across the crack area (white arrow). **h** Comparison of O K-edge and Mn L-edge EELS spectra collected (**g**). Relative oxygen concentration (**i**) and Mn L$_3$/L$_2$ edge intensity ratio (**j**) with the distance from the initial point in (**g**). The pink-shaded region indicates the crack region.

## Balancing the anionic/cationic redox for fatigue-resistant LLRO electrodes

Given the great benefits of anionic redox to exert high theoretical capacity and energy density, it is essential to activate as much oxygen redox as possible by increasing $y$ in LL$_y$ electrodes. However, as we demonstrated here, excessive utilization of oxygen-redox inevitably leads to asymmetric lattice evolution in a larger degree and the formation of microcracks, ultimately causing chemo-mechanical failure, even though the persistent voltage decay issues could be resolved for the use of the O2-type layered framework. This degradation mechanism and the trade-off between the capacity and the stability is quite analogous to what has been extensively discussed for conventional nickel-rich layered lithium oxides, which suffer from chemo-mechanical failure during repeated cycling as the nickel content increases[54–56]. The abrupt anisotropic collapse of the layered structure at high SOCs creates residual stress and subsequent intergranular microcracks at the secondary-particle level, exposing new surfaces that result in side reactions with electrolyte and exacerbating cycle stability issues. Nevertheless, it should be noted that the chemo-mechanical degradation in O2-LRLO electrodes can be substantially mitigated by a proper balance of anionic and cationic redox capabilities. As evidenced by the series of electrochemical tests shown in Fig.1d, e, the LL$_{0.15}$, LL$_{0.175}$ and LL$_{0.2}$ electrodes demonstrate exceptional cycle stability while primarily undergoing the O$^{2-}$/O$^-$ redox mechanism, which contributes to 61.0%, 69.4%, and 77.4% of the capacity, respectively. By effectively limiting the oxygen redox to below 77.4% through the addition of 14% Ni in the structure, asymmetric lattice variations are reduced to 1.8%. This significant decrease suppresses strain build-up and crack formation, ensuring exceptional fatigue resistance of the electrode particles. This engineering of the cationic/anionic redox balance results in a specific energy density retention of 660.9 Wh kg$^{-1}$ after 100 cycles for LL$_{0.15}$ electrode (Supplementary Fig. S14), highlighting its promise for further development. In light of these findings, it is clear that achieving a proper balance between anionic and cationic redox capabilities is crucial for maximizing the benefits of oxygen redox and minimizing the risk of chemo-mechanical failure in LRLO electrodes.

## Discussion

We introduced a series of oxygen-redox lithium-rich layered oxides for a high-energy-density cathode, which could retain the anionic redox participation as much as ~90% without noticeable voltage deterioration. While successfully maintaining a stable voltage owing to the O2-type structural reversibility in these electrodes, we observed that electrodes with higher anionic redox compositions experience greater lattice breathing in the layered framework particularly at high SOCs, leading to the formation of microcracks in electrode particles during repeated cycling. Based on our findings, we have established a trilateral relationship among anionic redox, microcrack formation, and electrochemical degradation. This relationship highlights the crucial need to balance the

anionic/cationic redox capabilities within the electrode. Our optimal O2-type LRLOs could substantially mitigate the asymmetric lattice breathing, suppress the chemo-mechanical degradation and retain the high discharge voltage of ~3.43 V (vs. Li/Li⁺) and the capacity of ~ 200 mAh g⁻¹, delivering an energy density of 660.9 Wh kg⁻¹ after 100 cycles. These results serve as a direct illumination of the double-edged characteristics of oxygen redox, providing essential guidance for the design of high-energy-density LRLO cathode materials. While we proposed a cation/anionic redox balance as one of the ways to resolve the complex deterioration behaviors, we can also learn from previous extensive studies to address the chemo-mechanical failure of nickel-rich layered materials, considering the mechanical similarity between the two systems. Various approaches such as grain-boundary engineering[43], thermal post-treatment[57], and/or certain metal doping[58] have been widely used to remedy the bulk failure mode of nickel-rich layered oxide cathodes. Moreover, given recent reports on partial oxygen redox participating in conventional layered cathodes operating at high voltages[59–61], our findings may provide new insights into revisiting high-capacity layered materials beyond LRLO.

## Methods

### Synthesis

All the corresponding P2-type sodium layered cathodes were synthesized using the sol-gel method. Starting from fully dissolving the resorcinol (99%, Sigma-Aldrich) and formaldehyde in deionized water as chelating agents, stoichiometric amounts of LiCH₃COO • 2H₂O (99%, Sigma-Aldrich), NaCH₃COO • 3H₂O (99%, Sigma-Aldrich), Ni(CH₃COO)₂ • 4H₂O (98%, Sigma-Aldrich), and Mn(CH₃COO)₂ • 4H₂O (99%, Sigma-Aldrich) were added to the mixture. To form a homogeneous gel, the solution was heated at 70°C under stirring for 2 h and dried at 90°C overnight. The yielded compounds were finely grinded and underwent heat treatment at 500°C for 5 h, followed by additional annealing at 850°C for 10 h after pelletizing to obtain the P2 phase. During the subsequent Na/Li ion-exchange process, the P2-phase powder was thoroughly mixed with 10 times excess amount of LiNO₃ and LiCl eutectic mixture and heated at 300°C for 5 h in air. After the exchange, the resultant powder was attained by rinsing the product with deionized water several times and drying in a 70°C vacuum oven.

### HRPD

HRPD analysis was conducted at beamline 9B of the Pohang Light Sources (PLS) at the Pohang Accelerator Laboratory (PAL), Republic of Korea. Diffraction patterns were collected over the 2θ range of 10°–130.5° with a step size of 0.01°, step time of 2 s, and wavelength of λ = 1.5216 Å.

### XANES

XANES spectra for the Ni and Mn K-edges of the electrodes were collected at beamline 7D and 8 C of the PLS. Two sets of Si(111) crystals were used as the double-crystal monochromator. All the spectra were obtained at room temperature; the Mn K-edge spectra were collected in transmission mode, whereas the fluorescence mode was used for Ni K-edge because of the relatively low Ni contents in the materials. To remove the higher-order harmonic signal contamination, the monochromator was detuned by 30% to reduce the incident X-ray intensity. The reference metal foils were placed in the intermediate chamber for accurate energy-scale calibration. The collected XANES spectra were analyzed using the ATHENA software package[62].

### Electrochemical tests

For the electrochemical tests, 80 wt% of active materials, 10 wt% of carbon black (Super P), and 10 wt% of polyvinylidene fluoride were dissolved in N-methyl-2-pyrrolidone (NMP; 99.5 %, Sigma-Aldrich) and cast onto aluminum foil. The slurry was dried in a 70°C vacuum

oven overnight. 2032-type coin cells (CR2032, Hohsen) were assembled for the lithium half-cells with the separator (GF/F, Whatman), a mixture of 1 M LiPF6 in ethyl carbonate and dimethyl carbonate (EC/DMC, 1/1 v./v.) as the electrolyte, and Li metal as the anode. All the processes were conducted in an Ar-filled glove box. Then the half-cells were cycled at a current density of 20 mA g⁻¹ in the voltage range of 2.0–4.6 V at 60 °C temperature, including the activation cycle.

### STXM

Scanning transmission X-ray microscopy (STXM) analysis for the O K-edge spectra was performed at beamline 10A1 of the PLS. The cycled electrodes were harvested and sonicated in DMC solvent to displace the active materials of the O2-LRLOs from the Al current collector. A few droplets of the suspension were cast on carbon-coated Cu TEM grids and dried in an Ar-filled glove box. To prevent unexpected contamination, the loaded samples were sealed under Ar and transferred to the beamline. The primary particles were placed in the focal plane for the measurements, and the transmitted beam intensity with specific energy was collected for each pixel of the 2D window. The image stacks were obtained in the energy range of 520.0–535.0 eV with varied step size (0.2–1 eV) and typically 1-ms dwell time and aligned in the aXis2000 software package.

### In situ XRD

In situ XRD measurements were performed to observe the structural changes of Li-rich cathode electrodes in the 2θ range of 10–40° during the charge/discharge process at 60 °C using coin-cell type homemade electrochemical cells with Li metal as the anode on PANalytical EXPERT pro diffractometer with Cu Kα (λ = 1.5406 Å) source. To maximize diffraction signal, thin Al layer (500 nm) was sputter coated on Be (125 μm) window and it acted as a current collector. The uniform slurry was directly casted on Al coated Be sheet with 14 mm circular mask and vacuum dried at 80 °C for 12 h. The Li rich electrodes loading mass was about 1.2 mg cm⁻². For electrochemical cycling, the current density was 20 mA g⁻¹ with potential window of 2.0–4.6 V. During the cycling process, X-ray diffraction patterns were recorded every 30 min using a PIXcel 1D detector (PANalytical).

### Raman spectroscopy

Raman spectra for the O2-LRLOs in various SOCs were obtained using a Raman spectrometer (LabRAM HV Evolution, HORIBA) at the Research Institute of Advanced Materials (RIAM), Seoul National University, Seoul, Republic of Korea with an Ar-ion laser as the excitation light source (λ = 532 nm). The objective lens magnification was X50 to focus the scattered Raman signal from the loaded electrodes. The data were collected with an acquisition time of 15 s of acquisition time and 10 times of accumulations. The peaks were deconvoluted by XPS Peak analyzer.

### DFT calculations

First-principles calculations for LL$_{0.25}$ using the projector augmented wave (PAW) method[63] were performed using the Vienna Ab Initio Simulation Package (VASP)[64], which is based on density functional theory (DFT). Structural relaxations were conducted within the spin-polarized generalized gradient approximation (GGA) with the Perdew–Burke–Ernzerhof (PBE) exchange-correlation functional[65]. The simulated compositions for LL$_{0.25}$ is Li(Li$_{0.25}$Ni$_{0.04}$Mn$_{0.71}$)O$_2$, which is slightly different from the experimental composition of Li$_x$(Li$_{0.25}$Ni$_{0.05}$Mn$_{0.7}$)O$_2$, but it is sufficient to describe the evolution trend of the lattice parameters in that they have comparable Li content and anionic redox activity. All the atomic positions were optimized until the interatomic forces converged to less than 0.02 eV Å⁻¹. A 520 eV of energy cut-off plane-wave basis set was adopted along with a 2 × 2 × 2 k-point grid based on the Monkhorst–Pack scheme[66], and

Python Materials Genomics (pymatgen) codes[67] were used to enumerate all the symmetrically distinct configurations for $LL_{0.25}$ in various SOCs with or without TM migration. During the enumeration process, 50 configurations with the lowest electrostatic energy for each composition were selected to perform a full DFT calculation, and finally, the configuration with the lowest energy was selected to calculate the lattice parameters, average magnetic moment, and TM migration energy (see Fig. 2j, Supplementary Note 4, and Supplementary Figs. S20–22).

## XNI

The 3D volumetric rendered images for O2-LRLOs were obtained from X-ray nano-imaging analysis, performed at beamline 7 C of the PLS. The 10th-charged electrodes were harvested and rinsed with DMC to remove the residual salts, and a piece of electrode was attached to the sample holder using optical microscopy. An X-ray energy of 9.3 keV was applied, and sequential transmitted 2D images from various directions were collected by rotating the samples by 180°. The corresponding 2D images were aligned and rendered into 3D images and video using the Avizo software package. The defect volume ratio of each particle was quantified by assigning the defect region for the whole slice images under the particular contrast threshold. The truncated mean was used to compare the defect ratio of O2-LRLOs to avoid outliers.

## Cs-STEM

Cross-sectional TEM specimens for the 10th-charged $LL_{0.25}$ were fabricated by focused-ion-beam milling (FIB, Helios G4) at the National Center for Inter-university Research Facilities (NCIRF), Seoul National University, Seoul, Republic of Korea. Under 200-keV electron beam energy, the HAADF images were collected with a point-to-point resolution of 0.08 nm using JEM-ARM200F and Themis Z, located at the RIAM and NCIRF, Seoul National University, Seoul, Republic of Korea, respectively. Corresponding electron-energy-loss spectroscopy (EELS) data for the Mn L-edge and O K-edge were analyzed with a 25 nm * 50 nm EELS window across the crack region, and the data were collected for every 1-nm detecting step. The EELS spectra were processed using Digital Micrograph (Version 3.43, GATAN).

## Data availability

All relevant experimental and computational data within the manuscript are available from the corresponding author upon reasonable request. Source data are provided in this paper. Source data are provided with this paper.

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

## Acknowledgements

This research was supported by the Basic Science Research Program through the National Research Foundation of Korea (NRF) funded by the Ministry of Education (2021R1A6A3A13039400, 2022R1A6A3A01086197, and 2021R1C1C2004527) and funded by the Korea government (MSIT) (No. RS-2023-00261543). This work was also supported by the Center for Nanoparticle Research at Institute for Basic Science (IBS) (IBS-R006-A2) and LG Energy Solution.

## Author contributions

H. J., D. E., and K. K. established the project idea and designed the experiments. H. J. and D. E. synthesized the materials and performed the electrochemical tests, Raman spectroscopy, and all the synchrotron-based HRPD, XANES, STXM, and XNI measurements. J. C. conducted the in situ XRD and analysis. J. L. and Y. L. helped interpret the XNI results. J. S., B. K. and D. K. performed and analyzed the DFT calculations and results. H. P. and J. H. provided advice for the experiments. S. C. and S. L. conducted STEM analyses and FIB milling. S. J. and J. W. L. provided the idea to analyze the synchrotron measurements. H. J., D. E., and K. K. wrote manuscript, and K. K. supervised the overall project.

## Competing interests

The authors declare no competing interests.
