## [Peer Review File · Nature Communications]

REVIEWER COMMENTS

Reviewer #1 (Remarks to the Author):

As a previous reviewer of this manuscript, I read the modified part and agree to accept this paper in current form

Reviewer #2 (Remarks to the Author):

This work investigates a series of Li-rich O2-type materials. Using X-ray nano-imaging, the authors observe intragranular cracking in their materials that increases in severity in the more Li-rich compositions with greater O-redox capacity. This is attributed to the larger and more asymmetric variation in c lattice parameter that they undergo during charging. This cracking appears to be correlated with poor capacity retention but not voltage fade. In line with other reports on O2-type cathodes, these materials do not show pronounced voltage fade over cycling compared to O3-type cathodes despite a significant contribution from oxygen redox.

The link between O2-type materials and reduced voltage fade has already been established in previous papers, including one by some of the present authors in Nature Materials. The origin of this lack of voltage fade has been attributed to greater reversibility of out-of-plane transition metal migration. The main new conclusion of the present work is that even in O2 materials there is a compromise to be found between gaining higher capacities from oxygen redox, at the cost of poorer capacity retention and worse mechanical failure.

This is not a surprising result given the context of understanding around oxygen redox, however, for the most part, this is a thorough study using appropriate techniques and data analysis and could be published in Nature Communications. However, there are some concerns that would need to be addressed before this paper could be considered publishable.

There is a significant doubt over the purity of the samples used in this study. The XRD in Fig S2 show several additional peaks not indexed to the P2 structure between 17-23 degrees and similarly in Fig S3 there appears to be a secondary layered phase present immediately to the right of the 002. These impurities should be identified and quantified with refinement as they are clearly present in a significant quantity. The impact of these impurities on the composition of the main phase and the rest of the data interpretation should also be discussed by the authors.

Cracking has been observed to occur during ion exchange due to strains induced from interlayer collapse when replacing the Na ions with Li (Karger et al. Chem Mater, 35, 2, 648-657). This interlayer collapse is more severe than that observed electrochemically. Can the authors confirm experimentally that the cracking they observe is not present in the as-prepared O2-type materials?

On the end of page 11, the authors write 'We could also observe that the transition metal migration, if it occurs, contributes to the lattice shrinkage additionally by approximately 0.24 Å as denoted with Δc_{TM} in the figure.' Interlayer contraction is not sufficient evidence for transition metal migration. Since it is critical to understanding oxygen redox, voltage fade and capacity retention behavior, can the authors provide experimental evidence whether transition metal migration occurs or not in their materials? Does the amount of transition metal in the interlayer correlate with the extent and asymmetry of c lattice change?

Reviewer #4 (Remarks to the Author):

I have gone through the comments made by the reviewers from the previous round of review at Nat. Mater. and the author's response letter. Overall speaking, this manuscript reports high-quality experimental data and insightful discussion on the O2-type cathode materials. It is certainly worthy of publication. I have some minor comments for the authors to consider.

(1) It may be necessary to provide STEM-HAADF images to verify the as-made materials are indeed O2-type materials. According to Fig S4, the as-made particles are actually quite large in size. I am wondering whether the ion-exchange reaction could be 100% complete. The authors might see some O3 or even P2 stacking in their materials, which may (or may not) have some impact on the electrochemical performance.

(2) Page 9, line 192, the authors claim "Furthermore, at a high current density of 1000 mA g⁻¹, LL0.15 and LL0.2 could retain 64.5 and 56.1% of their capacities at 20 mA g⁻¹, respectively, which is even superior to the reported rate performance of single-crystalline NCM cathodes (ref. 35) (Supplementary Fig. S12)." Strictly speaking, this is not a fair comparison. I am sure that the authors can find some other single-crystal NCM papers reporting outstanding rate performance. I guess the question that really matters to the battery community is that what would be the realistic solution to the poor kinetics of the O2-type cathode materials. The authors have been working on this family of cathode materials for several years now. They are in the position to provide a balanced and fair discussion or perspective.

(3) In the conclusion paragraph, the authors seem to suggest that conventional approaches developed for the NCM cathode materials may be useful for the O2-type cathode materials, which I respectfully disagree. The TM migration and anion redox in the O2-type cathode materials are happening in the bulk. Note that even doping in NCM is usually a surface doping (except for Mg and Al). I am wondering if the

authors may take the time to expand their discussion here. This would be greatly benefit the battery cathode research community.

Reviewer #1

General comments: *As a previous reviewer of this manuscript, I read the modified part and agree to accept this paper in current form.*

Our response: We sincerely appreciate the referee's positive feedback about our revised manuscript.

Reviewer #2

General comments: *This work investigates a series of Li-rich O2-type materials. Using X-ray nano-imaging, the authors observe intragranular cracking in their materials that increases in severity in the more Li-rich compositions with greater O-redox capacity. This is attributed to the larger and more asymmetric variation in c lattice parameter that they undergo during charging. This cracking appears to be correlated with poor capacity retention but not voltage fade. In line with other reports on O2-type cathodes, these materials do not show pronounced voltage fade over cycling compared to O3-type cathodes despite a significant contribution from oxygen redox.*

The link between O2-type materials and reduced voltage fade has already been established in previous papers, including one by some of the present authors in Nature Materials. The origin of this lack of voltage fade has been attributed to greater reversibility of out-of-plane transition metal migration. The main new conclusion of the present work is that even in O2 materials there is a compromise to be found between gaining higher capacities from oxygen redox, at the cost of poorer capacity retention and worse mechanical failure.

This is not a surprising result given the context of understanding around oxygen redox, however, for the most part, this is a thorough study using appropriate techniques and data analysis and could be published in Nature Communications. However, there are some concerns that would need to be addressed before this paper could be considered publishable.

Our response: We appreciate the referee's constructive comments. The main goal of our work was to establish the trilateral relationship among anionic redox capabilities, bulk microcrack formation, and electrochemical degradation in LRLO cathodes by adopting a compositional series of O2-LRLO cathode materials in which the irreversible transition metal migration and the voltage decay were effectively mitigated¹⁻⁴. In the following, we have provided detailed point-by-point responses to each comment.

Comment #1: *There is a significant doubt over the purity of the samples used in this study. The XRD in Fig S2 show several additional peaks not indexed to the P2 structure between 17-23 degrees and similarly in Fig S3 there appears to be a secondary layered phase present immediately to the right of the 002. These impurities should be identified and quantified with refinement as they are clearly present in a significant quantity. The impact of these impurities on the composition of the main phase and the rest of the data interpretation should also be discussed by the authors.*

Our response: As the referee pointed out, HRPD data of the P2 sodium phase in Fig. S2 did show minor peaks in the 17~23° two-theta range, and a similar peak at 18.6° was also observed even after the ion-exchange process. To identify the phase purity in a more clear and detailed manner, we magnified the 17~27° region of sodium P2 phases for LL_{0.15}, LL_{0.2}, and LL_{0.25} samples (P2-LL_{0.15}, P2-LL_{0.20} and P2-LL_{0.25}, respectively) in **Fig. R2-1**. According to previous studies⁵, the peaks at 20.2, 21.9, and 25.9° are generally attributed to the superstructure pattern arising from honeycomb orderings of lithium and TM ions in TM layer of P2 sodium layered phase. The peak intensities of these superstructure pattern typically increase as more LiMn₆ superstructures are present with higher Li and Mn contents, which is consistently observed in our case from P2-LL_{0.15} to P2-LL_{0.25}, supporting the idea that these are from the Li-TM ordering in TM layer. Nevertheless, we also found that Li₂MnO₃ impurity phase with a main peak at

18.6° was present in our P2-type sodium phases, which was more evident with higher-lithium-content P2-phases. These Li_2MnO_3 impurity peaks remained even after the ion-exchange in the O2-samples (Fig. S2).

Despite the presence of some impurity phase in $\text{LL}_{0.25}$, we justify that such impurity does not undermine the trilateral relationship we observed in O2-LRLO materials, and this is supported in several aspects:

1) The experimental composition determined from ICP analyses (Table. S1) closely matched with the expected values, while the portion of anionic redox was still correlated well with the Li contents in the structure (Figs. S5-6). Moreover, as illustrated in **Fig. R2-2**, DEMS analysis confirms the absence of any oxygen evolution even in the $\text{LL}_{0.25}$ which utilizes almost 90% of oxygen redox. If the remaining Li_2MnO_3 impurity affected the overall electrochemical activity of $\text{LL}_{0.25}$ in the prolonged cycles, $\text{LL}_{0.25}$ should have accompanied severe gas evolution at least during the first charging process^{6,7}. The negligible gas evolution in $\text{LL}_{0.25}$ thereby supports the trivial electrochemical impact of residual O3- Li_2MnO_3 impurity phase.

2) By closely monitoring the (003) peak of O3-phase impurity in the $\text{LL}_{0.25}$ through our *in-situ* and *ex-situ* XRD analyses (Figs. 2a-c in main text and **Fig. R2-3**), we confirmed that, unlike the main (002) peak of O2 structure, the main peak of the residual O3- Li_2MnO_3 impurity phase exhibited far imperceptible peak shift during the cycling, with $<0.08^\circ$ of 2θ . According to previous *in-situ* XRD study for conventional O3-type LRLOs, distinct peak shifts should be clearly observable towards lower 2θ region^{8,9} or even drastic lattice collapse at the end of charge state⁶. This discrepancy indicates that the electrochemical contribution of the O3-phase impurity in our LL_y materials is negligible, and the electrochemical performances presented in our manuscript should be predominantly attributed to the O2-phase.

We are now directing our efforts to improve the purity of the synthesized O2-phases by exploring alternative synthetic routes, such as co-precipitation or spray-pyrolysis methods.

Fig. R2-1. HRPD patterns for P2-LL_{0.15}, P2-LL_{0.2}, and P2-LL_{0.25} in 17 – 27° region and theoretical Bragg positions of O3-type Li₂MnO₃ impurity (navy) and superstructure patterns of P2-type sodium (cyan).

Fig. R2-2. Voltage profile and differential electrochemical mass spectrometry (DEMS) result of LL_{0.25} for 1.5 cycles.

Fig. R2-3. *Ex-situ* XRD patterns for LL0.25 in 2nd cycle. The red shaded region indicates the (003) peak of O3-phase.

Comment #2: *Cracking has been observed to occur during ion exchange due to strains induced from interlayer collapse when replacing the Na ions with Li (Karger et al. Chem Mater, 35, 2, 648-657). This interlayer collapse is more severe than that observed electrochemically. Can the authors confirm experimentally that the cracking they observe is not present in the as-prepared O2-type materials?*

Our response: We are grateful for the constructive comment. In order to verify whether the bulk micro-cracks were already present in the pristine materials or not, we conducted additional X-ray nano-imaging (XNI) experiments on as-prepared LL_{0.25} electrodes (**Fig. R2-4**), particularly as this material showed the most severe bulk structural degradation after cyclings. Our results from the figure show that LL_{0.25} in the pristine state displayed an intact bulk structure without appreciable crack volume portion (less than ~0.05%). This contrasts to ~ 7.9% of high crack volume ratio observed for the cycled LL_{0.25}.

Furthermore, by calculating the Full-width half-maximum (FWHM) of the main (002) peak of pristine LL_{ys} from the HRPD data in Fig. S3, we could confirm that our series of O2-LRLOs exhibit similar strain states in the as-prepared state (**Fig. R2-5**). Unlike the tendency of anionic redox capability or the degree of bulk degradation, the FWHM values have no clear correlation with the amount of Li_{TM}, indicating the comparable strain state in the pristine O2-LRLOs. Our series of additional experiments conclusively evidence that the bulk micro-cracks in the charged LL_{ys} are the result of electrochemical reactions and do not originate from the synthesis process.

Fig. R2-4. XNI results of as-prepared $LL_{0.25}$ electrode. **a**, XNI tomographic image of $LL_{0.25}$ electrode. **b**, Cross-sectional images of three-dimensional volume rendered image of the selected particle. **c**, Volumetric rendered images of ten $LL_{0.25}$ particles. Blue shaded region indicates the defective area in the particles.

Fig. R2-5. FWHM values of (002) peaks against the Li contents in TM layer.

Comment #3: *On the end of page 11, the authors write ‘We could also observe that the transition metal migration, if it occurs, contributes to the lattice shrinkage additionally by approximately 0.24 Å as denoted with Δc_{TM} in the figure.’ Interlayer contraction is not sufficient evidence for transition metal migration. Since it is critical to understanding oxygen redox, voltage fade and capacity retention behavior, can the authors provide experimental evidence whether transition metal migration occurs or not in their materials? Does the amount of transition metal in the interlayer correlate with the extent and asymmetry of c lattice change?*

Our response: Regarding the referee’s comment for evidence about the absence of TM disordering, we would like to underline that we had already provided *ex-situ* Raman spectroscopy for LL_{ys} in Fig. S13 to evidence the reversible TM back-migration after discharging process regardless of anionic redox capability. Raman spectroscopy is an effective tool that has been widely employed to detect the local phase transformation induced by TM_{Li} disordering in LRLO materials^{10,11}. It clearly illustrated that most of the TMs returned reversibly back to the TM layer after discharge, as estimated from the annihilation of A_{1g}^{defective} area, even after LL_{0.25} experiences a substantial amount of TM migration during de-lithiation process.

Fig. S13. a, Second-cycle Raman spectra for LLyS at every quarter of SOCs. **b,** Deconvolution of the peaks in the Raman shift range of 520–680 cm^{-1} of initial (bottom), fully charged (middle), and discharged (top) states for LLyS. **c,** Comparison of peak-area ratio between the spinel-like peak ($A_{1g}^{\text{defective}}$) and intact layered peaks ($A_{1g}+E_g$) at the end of charge (brown bar) and the degree to which the A_{1g} mode shifted to the left at 4.6 V (blue bar) for LLyS.

References

- 1 Zuo, Y. *et al.* A High-Capacity O2-Type Li-Rich Cathode Material with a Single-Layer Li₂MnO₃ Superstructure. *Adv. Mater.* **30**, 1707255 (2018).
- 2 Eum, D. *et al.* Voltage decay and redox asymmetry mitigation by reversible cation migration in lithium-rich layered oxide electrodes. *Nat. Mater.* **19**, 419-427 (2020).
- 3 Cui, C. *et al.* Structure and Interface Design Enable Stable Li-Rich Cathode. *J. Am. Chem. Soc.* **142**, 8918-8927 (2020).

- 4 Luo, D. *et al.* A Li-rich layered oxide cathode with negligible voltage decay. *Nat. Energy* (2023).
- 5 Saïbi, V. *et al.* Stacking Faults in an O2-type Cobalt-Free Lithium-Rich Layered Oxide: Mechanisms of the Ion Exchange Reaction and Lithium Electrochemical (De)Intercalation. *Chem. Mater.* **35**, 8540-8550 (2023).
- 6 Shen, S. *et al.* Tuning Electrochemical Properties of Li-Rich Layered Oxide Cathodes by Adjusting Co/Ni Ratios and Mechanism Investigation Using in situ X-ray Diffraction and Online Continuous Flow Differential Electrochemical Mass Spectrometry. *ACS Appl. Mater. Interfaces* **10**, 12666-12677 (2018).
- 7 Redel, K. *et al.* The impact of oxygen evolution and cation migration on the cycling stability of a Li-rich Li[Li_{0.2}Mn_{0.6}Ni_{0.1}Co_{0.1}]O₂ positive electrode. *J. Mater. Chem. A* **8**, 18143-18153 (2020).
- 8 Li, Q. *et al.* Improving the oxygen redox reversibility of Li-rich battery cathode materials via Coulombic repulsive interactions strategy. *Nat. Commun.* **13**, 1123 (2022).
- 9 Liu, T. *et al.* Origin of structural degradation in Li-rich layered oxide cathode. *Nature* **606**, 305-312 (2022).
- 10 Ruther, R. E., Callender, A. F., Zhou, H., Martha, S. K. & Nanda, J. Raman Microscopy of Lithium-Manganese-Rich Transition Metal Oxide Cathodes. *J. Electrochem. Soc.* **162**, A98 (2015).
- 11 Ku, K. *et al.* Suppression of Voltage Decay through Manganese Deactivation and Nickel Redox Buffering in High-Energy Layered Lithium-Rich Electrodes. *Adv. Energy Mater.* **8**, 1800606 (2018).

Reviewer #4

General comments: *I have gone through the comments made by the reviewers from the previous round of review at Nat. Mater. and the author's response letter. Overall speaking, this manuscript reports high-quality experimental data and insightful discussion on the O2-type cathode materials. It is certainly worthy of publication. I have some minor comments for the authors to consider.*

Our response: We sincerely appreciate the referee's thoughtful comments on our works. In response to the feedback, we have carefully provided detailed responses to each comment and made corresponding revisions to address the referee's concerns and clarify our manuscript.

Comment #1: *It may be necessary to provide STEM-HAADF images to verify the as-made materials are indeed O2-type materials. According to Fig S4, the as-made particles are actually quite large in size. I am wondering whether the ion-exchange reaction could be 100% complete. The authors might see some O3 or even P2 stacking in their materials, which may (or may not) have some impact on the electrochemical performance.*

Our response: The referee's concern about the phase purity of our series of LL_{ys} is valid, in light of the large particle sizes of pristine materials in Fig. S4. According to Inductively Coupled Plasma Spectrometry (ICP) results presented in Supplementary Table. S1, most Na ions in the P2 structure have been successfully replaced by Li ions in the O2 structure after the ion-exchange process. Furthermore, the absence of characteristic (002) peak of P2-phase at $2\theta \approx 16^\circ$ in O2-phase (Figs. S2-3) and the complementary STEM images in **Fig. R4-1** confirmed that our materials predominantly exhibit the O2-type stacking sequences. Nevertheless, it should be acknowledged that some impurity O3-phase ($2\theta \approx 18.6^\circ$ in Figs. S2-3) was detected both in the mother P2-phase and, consequently, in the ion-exchanged O2-phase, in which the extent of impurity phase slightly increased with the degree of Li-excess (**Fig. R4-2**). Given that

the ion-exchange temperature in our experiments is lower than the critical O2-O3 phase transition temperature (well above 400 °C)¹, the presence of O3-type impurities in O2-phase is likely stem from the synthetic process of the mother P2-phase. However, we justify that such phase impurity does not undermine the trilateral relationship we observed in O2-LRLO materials, and this is supported in several aspects:

1) The experimental composition determined from ICP analyses (Table. S1) closely matched with the expected values, while the portion of anionic redox was still correlated well with the Li contents in the structure (Figs. S5-6). Moreover, as illustrated in **Fig. R4-3**, DEMS analysis confirms the absence of any oxygen evolution even in the LL_{0.25} which utilizes almost 90% of oxygen redox. If the remaining impurity O3-phase affected the overall electrochemical activity of LL_{0.25} in the prolonged cycles, LL_{0.25} should have accompanied severe gas evolution at least during the first charging process^{2,3}. The negligible gas evolution in LL_{0.25} thereby supports the trivial electrochemical impact of residual impurity O3-phase.

2) By closely monitoring the (003) peak of O3-phase impurity in the LL_{0.25} through our *in-situ* and *ex-situ* XRD analyses (Figs. 2a-c in main text and **Fig. R4-3**), we confirmed that, unlike the main (002) peak of O2 structure, the main peak of the impurity O3-phase exhibited far imperceptible peak shift during the cycling, with <0.08° of 2θ. According to previous *in-situ* XRD study for conventional O3-type LRLOs, distinct peak shifts should be clearly observable towards lower 2θ region^{4,5} or even drastic lattice collapse at the end of charge state². This discrepancy indicates that the electrochemical contribution of the O3-phase impurity in our LL_y materials is negligible, and the electrochemical performances presented in our manuscript should be predominantly attributed to the O2-phase.

We are now directing our efforts to improve the purity of the synthesized O2-phases by exploring alternative synthetic routes, such as co-precipitation or spray-pyrolysis methods.

Fig. R4-1. a, HAADF- and b, ABF-STEM image of pristine LL_{0.25}. the red and violet spheres indicate oxygen and transition metal, respectively.

Fig. R4-2. HRPD patterns for P2-LL_{0.15}, P2-LL_{0.2}, and P2-LL_{0.25} in 17 – 27° region and theoretical Bragg positions of O3-type Li₂MnO₃ impurity (navy) and superstructure patterns of P2-type sodium (cyan).

Fig. R4-3. Voltage profile and differential electrochemical mass spectrometry (DEMS) result of LL_{0.25} for 1.5 cycles.

Fig. R4-4. *Ex-situ* XRD patterns for LL0.25 in 2nd cycle. The red shaded region indicates the (003) peak of O3-phase.

Comment #2: Page 9, line 192, the authors claim “Furthermore, at a high current density of 1000 mA g⁻¹, LL0.15 and LL0.2 could retain 64.5 and 56.1% of their capacities at 20 mA g⁻¹, respectively, which is even superior to the reported rate performance of single-crystalline NCM cathodes (ref. 35) (Supplementary Fig. S12).” Strictly speaking, this is not a fair comparison. I am sure that the authors can find some other single-crystal NCM papers reporting outstanding rate performance. I guess the question that really matters to the battery community is that what would be the realistic solution to the poor kinetics of the O2-type cathode materials. The authors have been working on this family of cathode materials for several years now. They are in the position to provide a balanced and fair discussion or perspective.

Our response: We are pleased to the referee's comment. Initially, we compared the rate capability of LL_{0.15} and LL_{0.2} with that of bare NCM single-crystalline cathodes^{6,7} to demonstrate the superior kinetic properties of our single-crystalline LRLO materials compared to NCM counterparts. However, considering the complex nature of anionic redox, direct comparison of rate capability between O2-type LRLOs and single-crystalline NCM cathodes may have some logical fallacy. Instead, we highlight that our materials still exhibit superior capacity retention at high current density compared to untreated polycrystalline O3-type LRLO materials⁸⁻¹⁰, and correspondingly revised the manuscript. Furthermore, as the referee mentioned, ongoing research is exploring several strategies to further enhance the kinetic properties of O2-LRLOs, including as particle size/morphology control or further compositional tuning.

Main manuscript

Original text (Page 9): Furthermore, at extremely high current density of 1000 mA g⁻¹, LL_{0.15} and LL_{0.2} retained 64.5 and 56.1% of their initial capacity, respectively, which is significantly higher values than other single-crystalline NCM cathodes³⁵ (Supplementary Fig. S12).

Fig. S12. Rate performances of LL_{0.15}, LL_{0.2}, and LL_{0.25}.

Revised text (Page 9): Furthermore, at extremely high current density of 1000 mA g⁻¹, the single-crystalline LL_{0.15} and LL_{0.2} retained 64.5 and 56.1% of their initial capacity, respectively, which is comparable or even superior to polycrystalline conventional O3-type LRLO materials³⁵⁻³⁷ (Supplementary Fig. S12).

Fig. S12. Rate performances of LL_{0.15}, LL_{0.2}, and LL_{0.25}.

Comment #3: *In the conclusion paragraph, the authors seem to suggest that conventional approaches developed for the NCM cathode materials may be useful for the O2-type cathode materials, which I respectfully disagree. The TM migration and anion redox in the O2-type cathode materials are happening in the bulk. Note that even doping in NCM is usually a surface doping (except for Mg and Al). I am wondering if the authors may take the time to expand their discussion here. This would be greatly benefit the battery cathode research community.*

Our response: We are grateful for the referee's valuable suggestion, and acknowledge the distinct nature of anionic redox in Li-rich layered systems. Our original statement in the conclusion paragraph aimed to highlight the commonality in terms of bulk electro-chemo-mechanical failure between O2-LRLO and NCM cathodes. We suggested that various strategies successfully applied to conventional NCM cathodes could also be suitable for improving our O2-LRLO materials. However, given our manuscript's primary focus on bulk degradation rather surface-related issues, we revised the suggested strategies to emphasize modifications and remedies more closely tied to bulk structural aspects, such as grain-boundary engineering¹¹, thermal post-treatment¹² and/or certain metal doping¹³.

Main manuscript

Original text (Page 16): Various approaches such as surface treatment⁴⁷, morphology control⁵², and metal doping⁵³ have been widely used to remedy the failure mode of nickel-rich layered oxide cathodes.

Revised text (Page 16): Various approaches such as grain-boundary engineering⁴³, thermal post-treatment⁵⁷, and/or certain metal doping⁵⁸ have been widely used to remedy the bulk failure mode of nickel-rich layered oxide cathodes.

References

- 1 Carlier, D. *et al.* On the metastable O2-type LiCoO₂. *Solid State Ionics* **144**, 263-276 (2001).
- 2 Shen, S. *et al.* Tuning Electrochemical Properties of Li-Rich Layered Oxide Cathodes by Adjusting Co/Ni Ratios and Mechanism Investigation Using in situ X-ray Diffraction and Online Continuous Flow Differential Electrochemical Mass Spectrometry. *ACS Appl. Mater. Interfaces* **10**, 12666-12677 (2018).
- 3 Redel, K. *et al.* The impact of oxygen evolution and cation migration on the cycling stability of a Li-rich Li[Li_{0.2}Mn_{0.6}Ni_{0.1}Co_{0.1}]O₂ positive electrode. *J. Mater. Chem. A* **8**, 18143-18153 (2020).
- 4 Li, Q. *et al.* Improving the oxygen redox reversibility of Li-rich battery cathode materials via Coulombic repulsive interactions strategy. *Nat. Commun.* **13**, 1123 (2022).
- 5 Liu, T. *et al.* Origin of structural degradation in Li-rich layered oxide cathode. *Nature* **606**, 305-312 (2022).
- 6 Zhang, F. *et al.* Surface regulation enables high stability of single-crystal lithium-ion cathodes at high voltage. *Nat. Commun.* **11**, 3050 (2020).
- 7 Liu, X. *et al.* Constructing a High-Energy and Durable Single-Crystal NCM811 Cathode for All-Solid-State Batteries by a Surface Engineering Strategy. *ACS Appl. Mater. Interfaces* **13**, 41669-41679 (2021).
- 8 Yu, X. *et al.* Understanding the Rate Capability of High-Energy-Density Li-Rich Layered Li_{1.2}Ni_{0.15}Co_{0.1}Mn_{0.55}O₂ Cathode Materials. *Adv. Energy Mater.* **4**, 1300950 (2014).
- 9 Kumar Nayak, P. *et al.* Remarkably Improved Electrochemical Performance of Li- and Mn-Rich Cathodes upon Substitution of Mn with Ni. *ACS Appl. Mater. Interfaces* **9**, 4309-4319 (2017).
- 10 Wu, J., Li, H., Liu, Y., Ye, Y. & Yang, Y. Doping and Coating Synergy to Improve the Rate Capability and Cycling Stability of Lithium-Rich Cathode Materials for Lithium-Ion Batteries. *J. Phys. Chem. C* **126**, 2410-2423 (2022).
- 11 Liu, X. *et al.* Origin and regulation of oxygen redox instability in high-voltage battery cathodes. *Nat. Energy* (2022).
- 12 Li, S. *et al.* Thermal-healing of lattice defects for high-energy single-crystalline battery

- cathodes. *Nat. Commun.* **13**, 704 (2022).
- 13 Ou, X. *et al.* Enabling high energy lithium metal batteries via single-crystal Ni-rich cathode material co-doping strategy. *Nat. Commun.* **13**, 2319 (2022).
-

REVIEWERS' COMMENTS

Reviewer #2 (Remarks to the Author):

The additional data and discussion provided by the authors is satisfactory. Provided the figures (R2-1 to R2-5) are all incorporated into a revised SI for final publication, this manuscript can be published.

Reviewer #4 (Remarks to the Author):

The authors have addressed my comments. I suggest that this manuscript may be considered for publication now.

Reviewer #2

General comments: *The additional data and discussion provided by the authors is satisfactory. Provided the figures (R2-1 to R2-5) are all incorporated into a revised SI for final publication, this manuscript can be published.*

Our response: We sincerely appreciate the referee's positive feedback about our revised manuscript. In response to the referee's valuable comment, we have properly incorporated additional supporting information as outlined below:

Main manuscript

Inserted text (Page 7): Despite the detection of minor O3-type Li_2MnO_3 impurities in both P2- and O2-structures (Supplementary Fig. S4), the marginal activity of the impurity phase (Supplementary Fig. S5-6) assures that the primary O2-phase dominates the electrochemical and structural responses.

Inserted Figures (Figs. S4-6)

Fig. S4. HRPD patterns for P2-LL_{0.15}, P2-LL_{0.2}, and P2-LL_{0.25} in 17 – 27° region and theoretical Bragg positions of O3-type Li₂MnO₃ impurity (navy) and superstructure patterns of P2-type sodium (cyan).

Fig. S5. Voltage profile and differential electrochemical mass spectrometry (DEMS) result of LL_{0.25} for 1.5 cycles.

Fig. S6. *Ex-situ* XRD patterns for $LL_{0.25}$ in 2nd cycle. The red shaded region indicates the (003) peak of O3-phase.

Original text (Page 14): It is contrast to the case of the pristine electrodes that were nearly absent with such structural defects (Supplementary Fig. S19)

Revised text (Page 14): It is contrast to the case of the pristine electrodes that were nearly absent with such structural defects and exhibited comparable stress states (Supplementary Figs. S23-24).

Inserted Figure (Fig. S23)

Fig. S24. FWHM values of (002) peaks against the Li contents in TM layer. No distinct correlation between peak broadness and Li contents in TM layer was observed, suggesting comparable bulk stress states among LL_ys.

Reviewer #4

General comments: *The authors have addressed my comments. I suggest that this manuscript may be considered for publication now.*

Our response: We are sincerely grateful the referee's positive feedback about our revised manuscript.